# Groundwater Vulnerability Analysis of Tirnavos Basin, Central Greece: An Application of RIVA Method

**Ioannis Vrouhakis [1,2,*]**, **Evangelos Tziritis [2]**, **Georgios Stamatis [1]** and **Andreas Panagopoulos [2]**

[1] Mineralogy and Geology Laboratory, Sector of Geological Sciences Department of Natural Resources & Agricultural Engineering, Agricultural University of Athens, Iera Odos 75, 11855 Athens, Greece; stamatis@aua.gr

[2] Soil & Water Resources Institute, Hellenic Agricultural Organisation "Demeter", Gorgopotamou Street, 57400 Thessaloniki, Greece; e.tziritis@swri.gr (E.T.); a.panagopoulos@swri.gr (A.P.)

\* Correspondence: i.vrouhakis@swri.gr; Tel.: +30-2310798790

**Abstract:** A novel index-based method (RIVA) for assessing intrinsic groundwater vulnerability was applied to Tirnavos basin (central Greece) to assess the susceptibility to surface-released contamination. Data from field surveys, previous studies, and literature were used to calculate the factors that compile the RIVA method. The aggregated results delineated the spatial distribution of groundwater vulnerability from very low to very high. The modelled results were successfully validated with ground-truth values of nitrates obtained from 43 boreholes. Overall, the modelled and the monitored values match more than 80%, indicating the successful application of the RIVA method. Few deviations were observed in areas dominantly affected by lateral crossflows and contamination from adjacent areas. RIVA proved an efficient method in terms of accuracy, data intensity, and investment to reach highly accurate results. Overall, RIVA proved to be a robust tool for reliable groundwater vulnerability assessments and could be further exploited for risk assessment and decision-making processes in the context of groundwater resource management.

**Keywords:** aquifer; groundwater; intrinsic vulnerability; RIVA method; index-overlay method; Tirnavos basin





## 1. Introduction

Groundwater accounts for nearly 99% of the total volume of freshwater presently circulating on our planet [1]. This overwhelming percentage, the lower susceptibility to pollution, and the large storage capacity of groundwater compared to surface water highlight its paramount importance at a global socio-economic level. Its significance, however, requires the most outstanding possible effort to protect it. A significant task towards this goal is the assessment of groundwater intrinsic vulnerability, a practice that suggests areas where priority and special attention should be given to the protection and overall management of groundwater resources. However, assessing the vulnerability of groundwater to adverse effects of human impacts is one of the most critical problems in applied hydrogeology [2]. The anthropogenic agricultural activities are often responsible for overdraft, groundwater quality deterioration, and increasing vulnerability. Due to level decline and quality degradation, sustainable development plans are needed to protect these resources [3].

In general, in most parts of the world, groundwater vulnerability assessment is based on (i) process-based methods [4–6], (ii) statistical methods [7–9], and (iii) overlay and index methods [10–13]. The limitations of process-based methods are adequate data and quality to capture the physical, chemical, and biological reactions from the surface to the uppermost aquifer. The statistical methods focus on the uncertainty by minimizing the error and using parameters' coefficients instead of weights. A possible drawback of these methods is the required monitoring data, which is essential. These methods are only applicable to those

regions where similar factors govern groundwater contamination. The overlay and index methods are the most suitable for groundwater vulnerability assessment, overcoming all the limitations mentioned above [14]. They focus on critical factors potentially controlling contaminant transport, and they are relatively cost-effective and adaptable to on-site specific conditions. Moreover, they demand minimum data to produce outcomes that can directly facilitate the decision-making processes [15]. The basic steps of these methods include raw data analysis, ranking of features on maps, integration of maps, and classification of the integrated map based on an index. These methods can be applied from a regional to global scale; nevertheless, they should be supplemented with field visits and on-site validation to produce reliable results [16].

Due to the advantages above and their ability to be easily used as strategic tools, the scientific community has increased interest. Some examples of this type of groundwater vulnerability assessment are DRASTIC (D: aquifer depth, R: recharge rate, A: aquifer lithology, S: soil type, T: topography, I: impact of vadose zone, C: aquifer hydraulic conductivity) [17], GOD (Groundwater occurrence, Overall lithology of aquifer, and Depth to groundwater level) [18], AVI (Aquifer Vulnerability Index) [19], SEEPAGE (System of Early Evaluation of Pollution Potential of Agricultural Groundwater Environments) [20], EPIK (Epikarst, Protective cover, Infiltration conditions, and Karst network development) [21], GALDIT (G: Groundwater occurrence, A: Aquifer hydraulic conductivity, L: Height of groundwater level, D: Distance from the shore, I: Impact of existing status of seawater intrusion, T: Thickness of the aquifer) [22], RISKE (Rock of aquifer media, Infiltration, Soil media, Karst, and Epikarst) [23], and a global risk approach [24].

Two types of aquifer vulnerability are considered. The first refers to intrinsic vulnerability, which considers the system's inherent properties, such as the properties of the vadose zone or the recharge conditions, etc. The second and more complex one is specific vulnerability. In addition to the intrinsic, it considers the properties of a specific contaminant or group of contaminants [25]. This research focuses on intrinsic vulnerability using a relatively new method (RIVA) [26]. The selected method is based on the successful concept of the European approach [27] and incorporates additional elements that provide more realistic and representative results. The main advantage of RIVA over similar methods is that it can be applied to all types of groundwater bodies independently of the specific conditions, lithologic phases, and aquifer typology of each area. This way allows uniform evaluation and comparison between hydrogeological systems with different characteristics. To this aim, the Tirnavos alluvial basin area is considered an appropriate test site for its application, as it includes an intensively cultivated area with diverse geological and hydrogeological characteristics. The proximity to the adjacent karstic system, combined with the occurrence of significant tectonic structures, further increases the aquifer system's vulnerability potential.

## 2. Case Study Area

Thessaly Plain in central Greece is the largest alluvial basin of the country and is divided, through the mid-Thessaly hills, into two sub-basins, the western Thessaly basin and the eastern Thessaly basin, developed in an NW-SE direction as part of the broader tectonic trough. The case study area where the RIVA method was applied is located in the northwest part of the wider eastern Thessaly basin (Figure 1). The hydrological perspective includes parts of the Titarisios River basin to the north and the Pinios River basin to the south. The total area is estimated to be 688 km$^2$, including the alluvial Tirvanos basin and part of its surrounding geological formations (Figure 2), which are hydrogeologically interdependent. The perimeter of the study area is 105 km, and the mean altitude is 160 m. The smallest and largest morphological gradients recorded are 0 and 62.57%, respectively, with a mean slope of 10.5%.

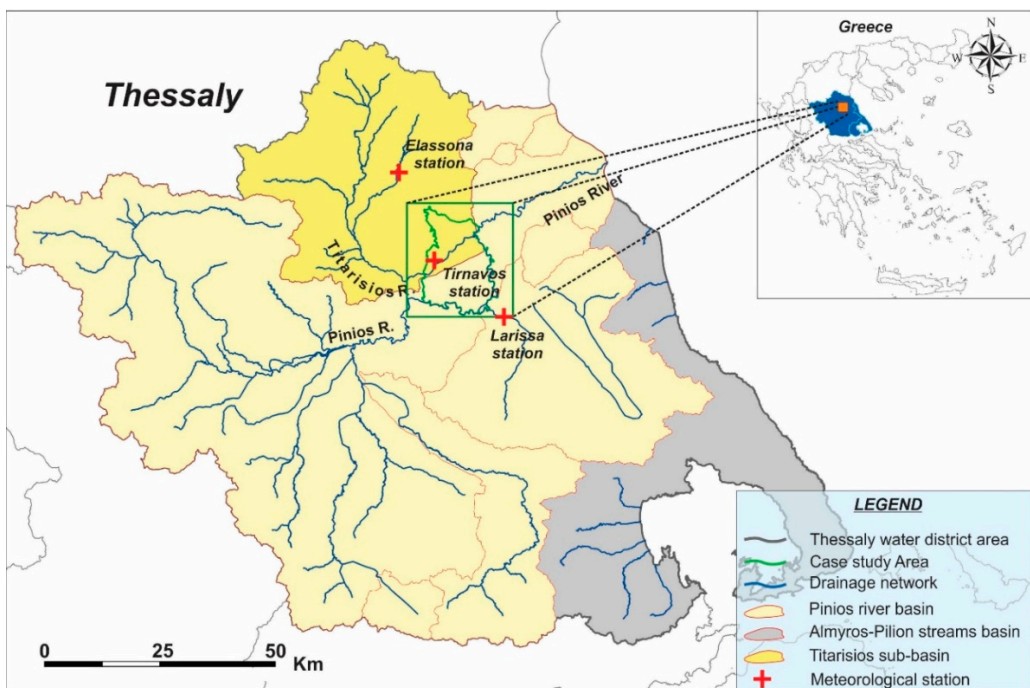

**Figure 1.** Location map of the case study area where the RIVA method has been applied.

The Tirnavos area is characterised by a typical Mediterranean climate, with annual rainfall from 400 mm to 600 mm, distributed almost entirely during the wet hydrological period, without significant summer precipitation. Larissa station is the closest one to the study area regarding the meteorological data, with continuous and reliable data over several years. Adjacent meteorological stations, used later in the methodology, are in Elassona and Tirnavos.

It is primarily a rural area covered by agricultural land, where intensified agricultural activities, both cultivation and livestock, are a significant source of groundwater contamination by nitrogen compounds. Manure waste and the often excessive and improper use of nitrogen fertilizers, aiming to improve agricultural production, lead to the occurrence of elevated concentrations of nitrates in groundwater [28]. Figure 3 shows the percentage distribution of land use according to CORINE categorization [29], suggesting that about 80% of land use consists of agricultural areas (arable land and permanent crops). The most common irrigation methods are drip irrigation and sprinkler, mainly using groundwater resources.

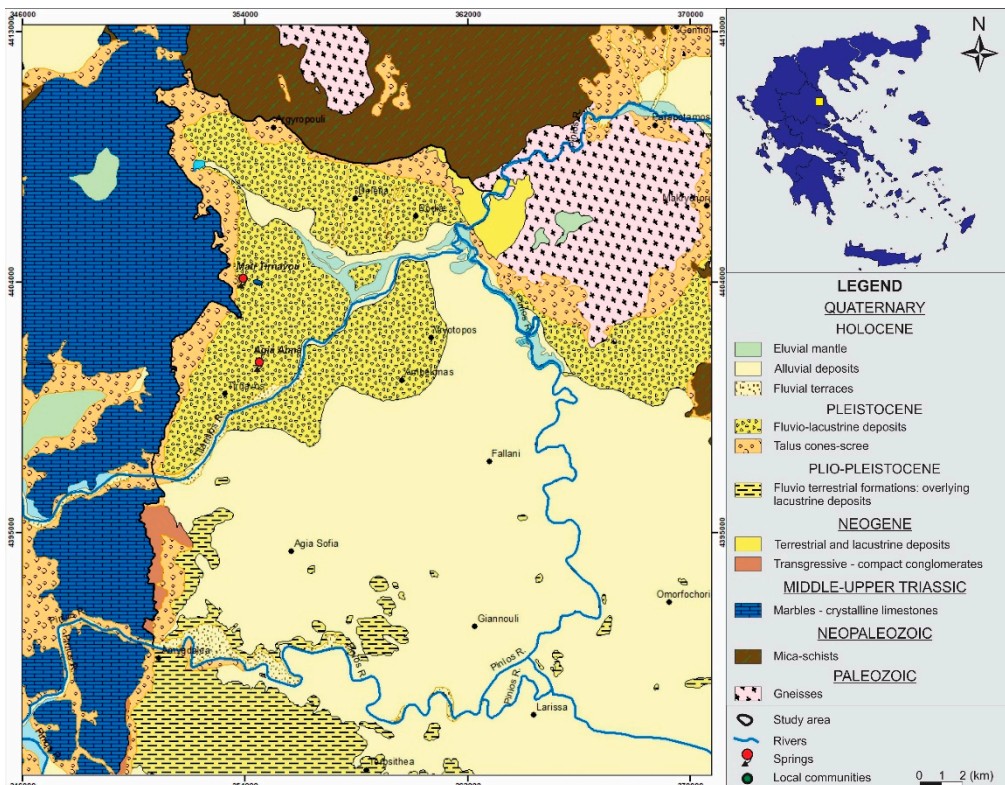

**Figure 2.** Geological map of the case study area, based on [30,31].

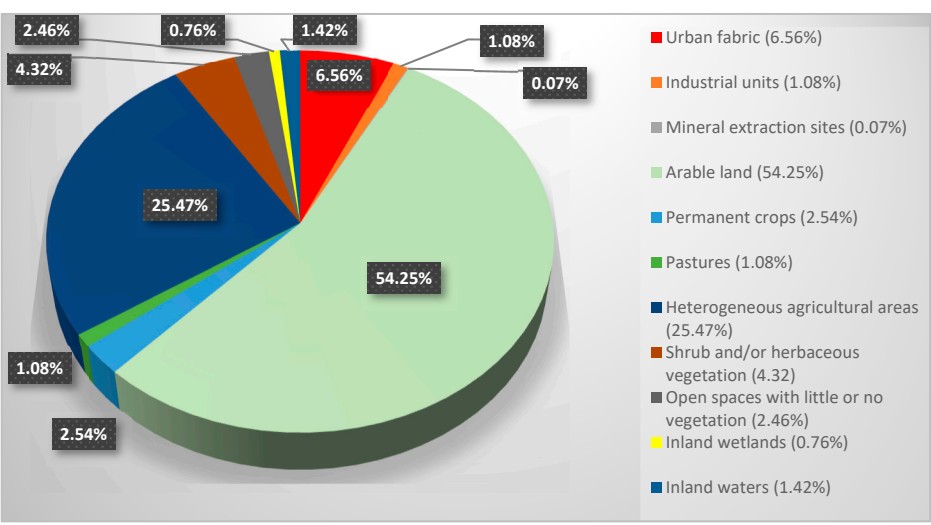

**Figure 3.** Land-uses distribution at the study area.

Three local irrigation organisations (Tirnavou, Ambelona, Agia Sofia) cater for irrigation water to producers through collective networks within their districts. Nowadays, all operating networks are pressurised, while until recently, an extensive gravitational network operated on the eastern part of the Ampelona district utilizing the karstic spring of Mati Tirnavou until 1998, when it was abandoned because of diminishing spring discharge. Despite the collective irrigation works, several privately owned wells exist and operate to cover irrigation demands of land that does not fall into the jurisdiction of the aforementioned collective networks, mainly in the northern and eastern ends of the study area [32].

Quaternary alluvial formations fill the boundaries along the southwest part of the basin with Neogene marls and sandy-clay deposits, whilst the western margins consist of

karstified marbles of the middle-upper Cretaceous. The crystalline bedrock is composed of mica-schists and gneisses of the upper Palaeozoic and Paleozoic age, respectively, forming the northern boundary of the basin (Figure 2). Two major springs (Mati Tirnavou and Agia Anna, Figure 2) emerge at the contact of the karstified system with the alluvial deposits. Pinios and Titarisios rivers flow across the basin, which, as already mentioned, hydrologically is part of the wider Pinios River basin.

The Quaternary deposits host an unconfined aquifer near the talus cone of Titarisios at the northwest, which, towards the central parts of the basin, sinks under a sequence of clay layers that form an aquitard [32–34] and have a maximum thickness of over 550 m at their central parts [32,35]. Under this setup, confining conditions occur, while the phreatic aquifer has been seriously affected due to systematic over-abstraction and is almost depleted [28,33]. The marbles at the western margins of the basin host a karstic aquifer of great potential, which recharges the alluvial system by lateral crossflows [28,33,34,36].

In the western part of the basin, an extensive marginal cone exists, through which the alluvial system receives significant amounts of recharge as crossflow through the Titarisios River gorge sediments. To the south, a smaller volume recharges the aquifer system as crossflow from the Pinios River gorge sediments [32].

Crossflow from the crystalline bedrock at the northern margins of the basin also occurs but is of minor importance. The southern extent of the karst system and from the mid-Thessaly hills recharge the central plain parts by crossflows from the southwest and the southern part of the area; however, they have a lower potential than that of the northern parts due to the existence of marls.

Based on the above data, it is perceived that the specific area is characterised by increased complexity, as reflected by the geological and hydrogeological conditions, tectonics, hydrodynamics, and land and water use. In addition, the area lacks regional planning and management of natural resources, leading to considerable deterioration of existing groundwater reserves. For this reason, RIVA was selected as an appropriate method of intrinsic vulnerability assessment to be implemented in the Tirnavos basin.

## 3. Materials and Methods

RIVA is the acronym of the four key factors considered in assessing intrinsic vulnerability, each of which has a distinguished control over groundwater vulnerability. The first factor is the recharge (R), which refers to the overall assessment of the effect of recharge conditions. The second one is the infiltration (I factor), which considers the soil's infiltration, subsequently affecting deep percolation to the saturated zone. The third factor relates to the protective cover and specifically to the impact of the vadose zone (V factor). The last factor is focused on the aquifer characteristics (A factor) and relates to the potential contaminant migration within the saturated zone as assessed by the hydraulic conductivity of the aquifer. Each of the four factors affects groundwater vulnerability individually and independently of the potential interactions. The final assessment of each factor is related to five classes of vulnerability (Figure 4), from very low (VL) to very high (VH), respectively [26]. The cumulative effect of these four factors constitutes the final assessment of intrinsic groundwater vulnerability, according to the relationship

$$i = R + I + V + A \tag{1}$$

The contribution of each factor in Equation (1) is not equivalent due to the different degrees of importance. Hence, each one is multiplied by a weighting factor, which reflects its significance to the result. The weighting factors (a, b, c, d), which were calculated with the use of Analytic Hierarchy Process (AHP) and proposed by [26], are a = 0.40 for V factor, recognizing that is the most important, followed by the I factor with b = 0.30, c = 0.15 for R factor, and d = 0.15 for A factor (a + b + c + d = 1). V factor is the most important one in shaping the overall vulnerability, as it assesses the buffering capacity of the geological medium in constraining deep percolation of the potential pollutants to the saturated zone, the flux of which is directly related to their leaching rate potential as defined by factor I,

which holds the second highest weight. Weighting factors' values are thus deduced as a fusion of the scientific approaches and documented viewpoints proposed by a number of researchers in the international literature [26].

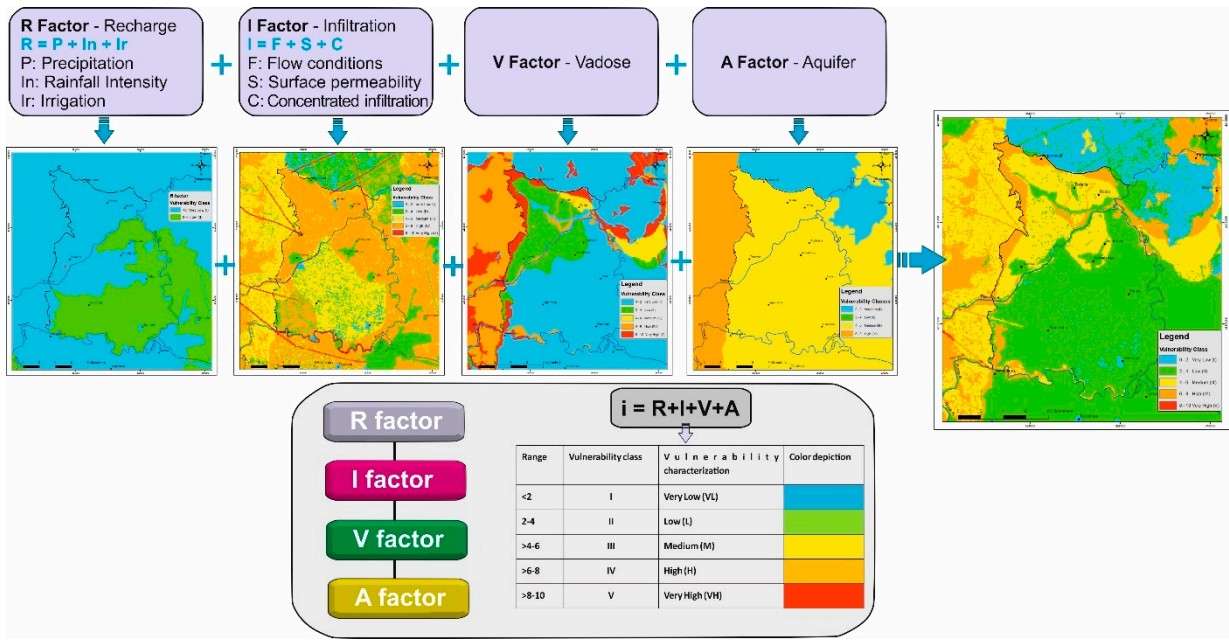

**Figure 4.** Schematic presentation of RIVA method (according to [26]).

In Table 1, the data used for the calculation of each factor and sub-factor of the RIVA method as well as their source of origin are presented. The tables of each method's classes and ratings can be downloaded from the link at the end of this paper as a supplementary document. The analytical description of RIVA's rationale and conceptualization per factor is beyond the scope of this research and may be reached through the original application of the method [26].

**Table 1.** Aggregate data used for RIVA method implementation in Tirnavos basin.

| Factor | Sub-Factor | Material | Source |
|--------|-----------|----------|--------|
| R | P | Rainfall data series | [37,38] |
| | In | Rainfall data series | [37,38] |
| | Ir | Spatial data of irrigated fields | [39] |
| | | Dominant irrigation methods | Local Land Reclamations Organizations (Tirnavou, Agia Sofia, Ambelona) personal communication |
| I | F | Digital Elevation Model | [40] |
| | | Land use | [39] |
| | S | Soil map | [41] |
| | | Degree of karstification | [28,32] |
| | | Geological map | [30,31] |
| | C | Main fault zones | [42,43] |
| | | Hydraulic interactions | [28,32] |

**Table 1.** *Cont.*

| Factor | Sub-Factor | Material | Source |
|:---:|:---:|:---:|:---:|
| V | - | Piezometric data (alluvial) | [28,34,36] |
| | | Bottom of confining aquifer | [44] |
| | | Piezometric data (karst) | [45] |
| | | Metamorphic formations weathered mantle | [28] |
| A | - | Hydraulic conductivity (Alluvial aquifer) | [33,44] |
| | | Hydraulic conductivity (Marbles, Gneisses, Schists) | [46–49] |

## 4. Results

### *4.1. R Factor Description*

R factor refers to the assessment of the internal vulnerability of the groundwater body as a function of the total surface recharge it receives and which can reach the aquifer through percolation under certain conditions. It includes three sub-factors [26], which are expressed through Equation (2):

$$R = P + In + Ir \tag{2}$$

where

R = recharge factor
P = precipitation sub-factor
In = rainfall intensity sub-factor
Ir = irrigation recharge sub-factor

As in the case of Equation (1), the contribution of each factor in Equation (2) is not equivalent. The adopted weights are 0.5, 0.3, and 0.2 for P, Ir, and In, respectively, which are embedded in the intermediate calculations of each factor and are not shown in the initial equation. The cumulative effect of these three sub-factors constitutes the final assessment of the R-factor (Equation (2)), which is linked to the intrinsic vulnerability of the groundwater system according to the classification shown in Figure 4.

#### 4.1.1. P Sub-Factor Calculation

Sub-factor P refers to the total precipitation depth received at the examined area over a year (y), in millimetres (mm). The representative precipitation value is deduced as the mean annual value over a sufficient time, e.g., over 20 years. If more than one station is available, the final figure is calculated by spatially integrating precipitation distribution of individual stations over the examined area, employing an appropriate interpolation scheme. The corresponding P factor values concerning precipitation ranges, according to the authors [26], are as illustrated in Table S1 in the supplementary documents.

In the case study area, there are three mereological stations (Figure 2) that were taken into consideration in the P sub-factor calculations: Larissa and Tirnavos stations located within the examined area, at 74- and 92-m altitude, respectively, and Elassona station at 314-m altitude, being representative of the higher elevation parts included in the examined area. For each station, precipitation data for a period of 30 years (1989–2018) were used to obtain a mean annual precipitation value for each of them. The calculated values were 422.2, 485.7, and 520 mm for Larissa, Tirnavos, and Elassona stations. Based on class values tabulated in Table S1, a P value of 2.5 is assigned to all three stations, and therefore, no spatial distribution is required.

4.1.2. In Sub-Factor Calculation

Except for the total precipitation, aquifer recharge depends on the intensity of the precipitation, which represents the amount of water received in a specific period. The effect of this parameter is difficult to estimate, as various factors, such as soil texture, root system, moisture state of the soil, the discontinuities of the geological formations, etc., form different conditions [50]. Nevertheless, it is generally accepted that an increase in rainfall intensity eventually increases the recharge and consequently the infiltration [51,52], thus affecting the vulnerability of the groundwater system. Regarding its calculation, the approach of the RIVA method is a compilation of approaches followed in similar methods [53,54]. The In values are calculated through Equation (3) [26]:

$$In = \Sigma P / \Sigma d \tag{3}$$

where

In = intensity value of rainfall
$\Sigma P$ = total precipitation at a given period (mm)
$\Sigma d$ = total number of rainfall days in the same period (days)

The rain intensity values for all stations were in the same range according to Table S2 in the supplementary documents (5.8 for Larissa, 8.9 for Tirnavos, and 9.5 for Elassona). Hence, the same In value (0.4) and vulnerability class (II) are assigned to the stations, and in this case, no spatial distribution method needs to be applied.

4.1.3. Ir Sub-Factor Calculation

The Ir sub-factor represents the effect of irrigation on groundwater vulnerability and constitutes a parameter that has not been considered in vulnerability by any other method. This is possibly due to the difficulty of estimating the volume of irrigation water used and its spatial distribution, as several variables are included that are not easy to quantify. However, the importance of the irrigation effect in groundwater vulnerability is recognized, especially in intensively cultivated regions, such as the examined area. Hence, RIVA, considering the irrigation impact through a qualitative approach based on the appraisal of the mean overall performance, as shown in Table S3 (Supplementary document) [26], is deemed appropriate for assessing vulnerability in the examined area.

The irrigated fields at the case study area are spatially distinguished based on data from the Greek Payment Authority of Common Agricultural Policy Aid Schemes [39]. For the categorization of the irrigated fields in accordance to the classification referenced in Table S3, data retrieved from the Local Land Reclamation Organizations of the region were used along with assessments carried out during in situ visits to the area, accounting also for the irrigation methods employed in each part of the basin, in cases where no specific data were available. Thus, drip-irrigated fields were assumed to apply nominal irrigation doses and result in minimal leaching rates due to the high efficiency the method. This is in contrast to sprinkler irrigation, where due to low water use efficiency, significant water losses occur that lead to considerable leaching rates, and therefore, increased irrigation doses are applied to satisfy crop water demands.

Based on this spatial data distribution according to the classes discussed (Table S3), the Ir sub-factor spatial distribution map was extracted, as illustrated in Figure 5.

*4.2. I Factor Description*

The I factor refers to assessing the intrinsic vulnerability of a groundwater body as a function of the surface infiltration conditions upon which deep percolation to the geological layers and eventually groundwater recharge depends. Its calculation is based on Equation (4), and its classification is shown in Table S1.

$$I = F + S + C \tag{4}$$

where

I = infiltration factor
F = flow conditions sub-factor
S = permeability of the surface medium
C = concentrated infiltration sub-factor

In Equation (4), F and S sub-factors have equal weighting factors (0.5), while sub-factor C does not contribute equally. Sub-factor C refers to the concentrated flow that could cause increased infiltration due to specific surface structures, such as epikarst, river beds, and tectonic contact. It is considered a critical aspect, which may induce maximum vulnerability to the groundwater body if existing.

### 4.2.1. F Sub-Factor Calculation

The F sub-factor relates to the surface water flow conditions, which affect infiltration to the saturated zone and consequently vulnerability of the groundwater system. Its calculation is determined by two parameters, the topographic slope (s) and vegetation (v). Based on the method's developers [26], the classification of "s" is made according to the approaches of [55–57], as shown in Table S4 in the supplementary documents.

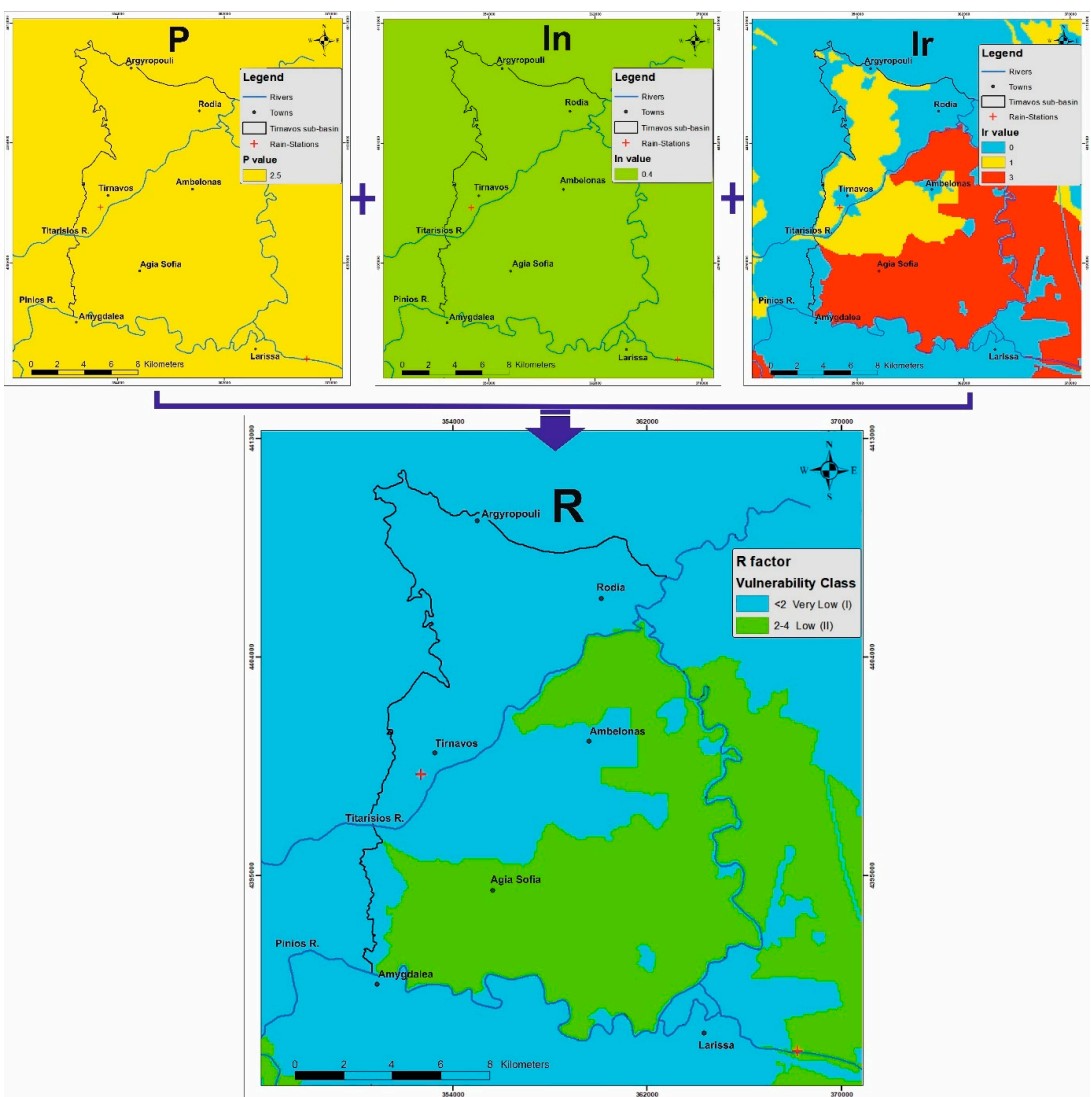

**Figure 5.** Spatial distribution maps of P, In, and Ir sub-factors in the Tirnavos basin and the R factor map compiled as a synthesis of the sub-factors above.

The vegetation (v) parameter is used as a slope correction parameter, as the denser it is, the more it includes plants with a well-developed root system, the more the surface runoff is prevented, and the more the surface infiltration is favoured, creating conditions of higher vulnerability. Vegetation is classified into three general groups: (a) forest areas (high vegetation), (b) cultivated and grassland areas (low vegetation), and (c) absence of vegetation or sparse vegetation.

Classification of vegetation was made according to the widely acceptable CORINE categorization [29] and shown in Table S5 (Supplementary document), where only Categories 2 (agricultural areas) and 3 (forests and semi-natural areas) related to vegetation are considered relevant. Based on the method, for the rest of the CORINE categories, i.e., 1 (artificial areas-artificial surfaces), 4 (wetlands), and 5 (water bodies), it is assumed that no surface flow occurs; thus, by definition, the sub-factor F takes the value zero (0).

Based on older studies [58,59], the effect of the vegetation parameter (v) was assessed comparatively for all considered land covers; regarding forest vegetation (forest), the latter assumed to result in the least total soil loss and runoff. High-vegetation areas (forest) may cause a decrease of surface runoff coefficient up to 88%, while low-vegetation areas (pastures) cause up to 44% decrease, respectively. The result constitutes the final vulnerability class of the sub-factor F, which takes values from 1 (very low vulnerability—V.L.) to 5 (very high vulnerability—V.H.) (Table S6, Supplementary document). Therefore, the final value of the sub-factor F to be used in Equation (4) will be deduced empirically from the combination of slope parameter (s) and vegetation parameter (v), according to Table S6 in the supplementary documents.

Based on the slope (s) parameter distribution (Table S4) as derived by the digital elevation model of the study area [40] and the vegetation (v) spatial distribution in accordance to the classes discussed (Table S5), the F sub-factor spatial distribution values were calculated regarding Table S6, as illustrated in Figure 6.

### 4.2.2. S Sub-Factor Calculation

The S sub-factor accounts for the permeability of surface geological formations and is directly proportional to the vulnerability of groundwater systems. The higher the permeability, the greater the vertical infiltration (percolation) and therefore the more significant recharge that will potentially reach the saturated zone of the groundwater system. RIVA regards surface formations that occur up to 1.5 m below surface and control surface/subsurface flow [26].

Surface formations are classified into soils and consolidated geological formations. Soils are considered the upper loose earth horizons, including topsoil, developed over non-consolidated geological formations (e.g., Neogene formations, Quaternary deposits). As consolidated geological formations, RIVA considers those that are lithified and constitute the underlying bedrock. The method assumes that soils of considerable thickness are not developed over the consolidated geological formations. Therefore, the surface hydrological conditions are controlled mainly by the permeability of the consolidated formations and not by the soil compartment.

S values for soils are shown in Table S7 (Supplementary document) based on the U.S. Department of Agriculture classification that is based on their texture [60], whereas, for the geological formations, S values are deduced based on their permeability, as proposed by the British Geological Survey and presented in Table S8 (Supplementary document) [61]. Permeability is a property that is not easily determined accurately, as it is affected by several factors (e.g., degree of fracturing, tectonic stress, karstification, alternations with horizons of different permeability, homogeneity, isotropy, etc.), which may more or less change the original nature of the formation. However, guidelines for a more general characterization framework are not limited and provide a range of values, as shown in Table S8. The lower values correspond to solid (unaffected) formations, while the higher reflect the factors above effect that eventually increases their permeability and thus their vulnerability class. The final calculation of the S sub-factor and its spatial distribution map results from compiling

F values for the soils and F values for the consolidated geological formations. Note that for each cell of the generated map, a value S of soil or formation is assigned and not an aggregated value of two individual values of soil and formation.

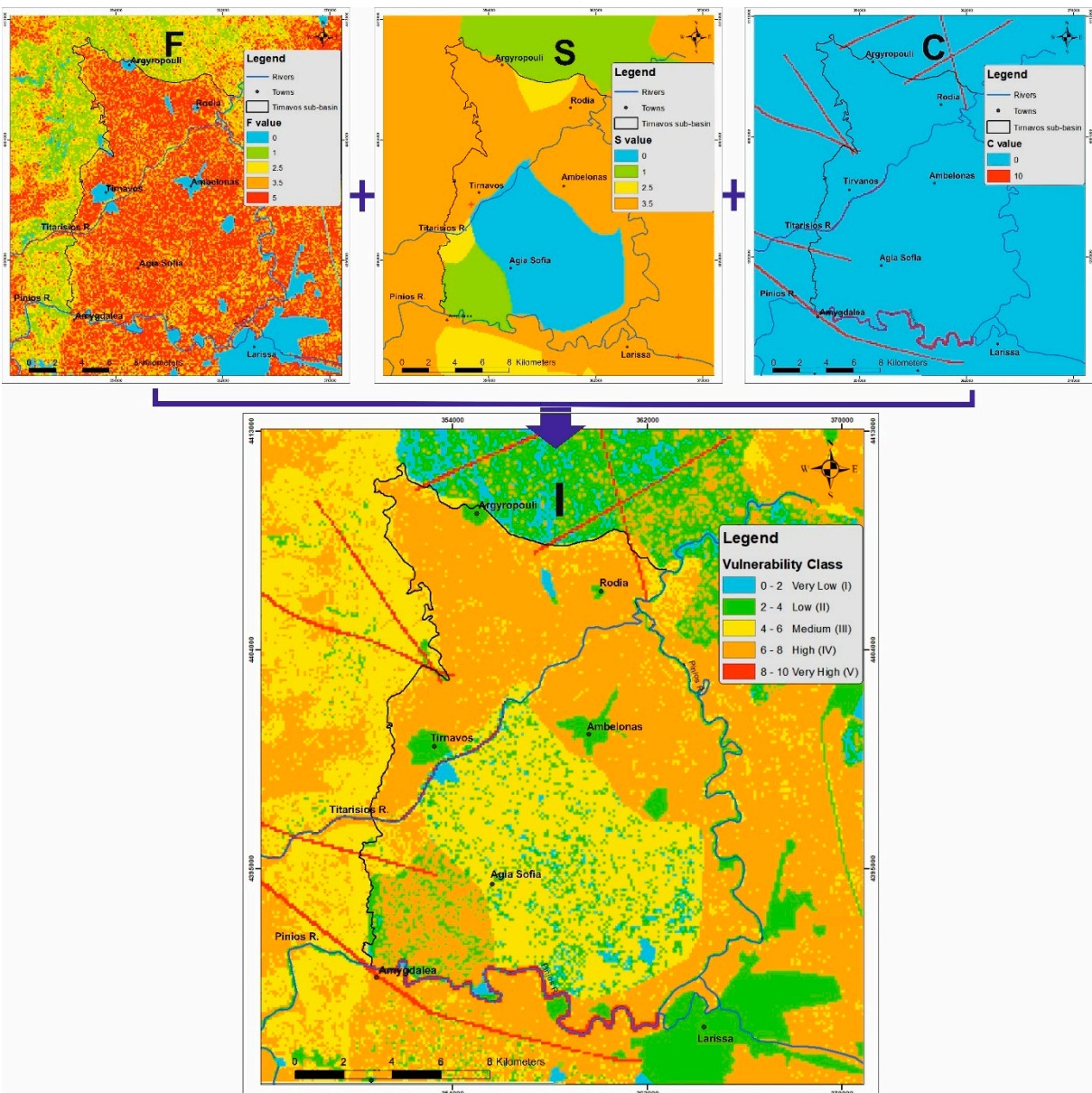

**Figure 6.** Spatial distribution maps of F, S, and C sub-factors in the Tirnavos basin. According to the I factor, the final map of vulnerability classes results from the composition of the three above mentioned thematic maps.

Regarding the Tirnavos basin, Quaternary deposits based on the geological map (Figure 1) were classified based on Table S7 categories, considering the soil study of the area [41] in which soil units were assigned with soil and geological and topographic criteria. In addition, Neogene formations at southwest and northeast margins and marbles at the west and the crystalline bedrock, composed of mica-schists and gneisses at the northern boundary of the basin, are classified based on Table S8. In addition, taking into account the degree of karstification of the marbles [28,32,34], vulnerability class III is assigned to them, and class II is assigned to the Neogene marls (resulting from their permeability based on the same studies). The above data processing composed the distribution map of the S sub-factor, as shown in Figure 6.

### 4.2.3. C Sub-Factor Calculation

Sub-factor C addresses the special cases of concentrated flow in fractured/discontinuities media that may impact the vulnerability characterization attributing the maximum vulnerability (very high) value to factor I. This sub-factor refers to the spatially concentrated because of specific surface features, which results in increased infiltration and thus maximum aquifer vulnerability. These features may include (a) epikarst; (b) drainage patterns, which are documented, that is, in hydraulic relation with the aquifer; (c) sinkholes; and (d) tectonic structures (e.g., faults, overthrusts). Infiltration is significantly favoured within the influence zone of the above features, causing eventually very high vulnerability.

As the exact orientation of the impact zones is not possible to be defined due to variable influencing factors that require complicated modelling approaches, RIVA proposes using an approximate impact zone of 100 m around critical surface features, to which a maximum score of 10 (V.H. vulnerability) is attributed [26]. If such structures do not occur (therefore no influence zones), the C value is by definition negligible. Hence, the C sub-factor constitutes an on-off (0–10) feature.

In the case of the Tirnavos basin, there are no indications of the existence of developed epikarst or sinkholes. However, some significant fault zones [42,43] can increase the infiltration and, consequently, the aquifer vulnerability (C values map, Figure 6). Furthermore, in earlier studies [28,32], it is suggested that in certain parts of the Pinios and Titarisios riverbeds, the hydraulic relationship between rivers and the groundwater system occurs. Regarding the Pinios River, confirmed hydraulic relation exists from its entrance to the Tirnavos basin up to Larissa city. In contrast, for the Titarisios River, significant hydraulic interaction occurs along the western margins of the basin, where extensive talus cones are formed (C values map, Figure 6).

### 4.3. V Factor Description

Factor V (vadose zone) accounts for the protection provided by the vadose zone as a function of the nature of its geological formations and its total thickness, which is directly related to the piezometric level. Factor V takes values from I (very low vulnerability) to V (very high vulnerability) according to the grading of Figure 4. As mentioned, it differentiates from the I factor because it regards the part below 1.5 m from the surface (upper soil horizons). In this context, the V factor may include (a) the soil's underlying non-lithified geological formations and/or strongly weathered zones of bedrock and (b) the bedrock (lithified geological formations).

Calculation of the V factor is performed through the modification and compilation of previous approaches [14,61–63] as follows [26]:

1.  Initially, each geological formation of the vadose zone of the study area is classified according to its dominant lithological type, prior to any secondary effects (e.g., karstification), and is attributed a "reference layer (ly) value" based on its permeability range, as shown in Table S9 in the supplementary documents.
2.  The "ly" values are multiplied by the fracturing or karstification factor (f), corresponding to an internal modification of the initial value "ly" due to secondary effects that impact permeability. The "f" factor derives from assessing the fracturing/karstification degree of the considered geological formation (only for the lithified) based on the values of Table S10 in the supplementary documents.
3.  The derived product is multiplied by the total thickness of the formation in meters to provide the final value of the protective cover (pc), which corresponds to a class of V factor, shown in Table S11 (Supplementary document). If the vadose zone consists of more than one layer in the vertical dimension, each formation is calculated individually as described, and then, all are summed up to calculate the final "pc" value.

From the above methodology, the critical point is to estimate the thickness of the formation (or formations) that shape the vadose zone. For this purpose, the following approach was adopted. For the phreatic aquifer zone within the alluvial basin, the piezometric level distribution as deduced from field measurements was considered [28,34,36], along

with the compiled digital elevation model (DEM) (subtracting piezometric map contours from DEM). When only a confined aquifer is active, the thickness of the vadose zone is based on the bottom of the confining layer as mapped by older foundation studies in the region [44], subtracted from the DEM. For the karstic domain considered, the piezometric levels provided by older studies were considered [45], under the assumption that no significant changes have occurred in the spatial distribution of the groundwater levels in this environment. If they have occurred, they would have related to groundwater level decline, leading to an increased thickness vadose zone and consequently lower vulnerability assessment. Therefore, this assumption provides a rather conservative calculation, favouring the area's environmental protection. No piezometric data exist for the crystalline bedrock at the northern boundary of the basin. However, a low-capacity aquifer is reported to occur in the weathering mantle of the metamorphic formations, the thickness of which does not exceed 30 meters [28]. Therefore, it is reasonable to assume that groundwater level will not occur at a depth greater than that. Hence, this value is used as a baseline for the vadose zone thickness in this environment.

The compiled map for vadose zone thickness, based on the above data, was multiplied by the map synthesized as the product of ly and f (ly × f), based on Tables S9 and S10, and the geological map of the area, in order to deduce a protection cover (pc) map.

### 4.4. A Factor Description

The A factor (aquifer) refers to the easiness with which a potential contaminant will travel within the saturated zone of an aquifer as a function of its hydraulic conductivity. Factor A takes values from 1 for an aquifer of very low hydraulic conductivity (very low vulnerability—V.L.) to 5 for an aquifer of very high hydraulic conductivity (very high—V.H.) according to the classification in Figure 4.

The value of factor A is calculated based on the quantitative or qualitative assessment of the hydraulic conductivity of the aquifer. In the case of measured values, the link between hydraulic conductivity and vulnerability class (A factor values) is shown in Table S12 in the supplementary documents, based on a modified, generic, yet widely accepted conductivity classification according to [64].

For the alluvial basin, the values of the hydraulic conductivity of the geological formations were derived based on the results from the pumping test analyses [33,44] and the geo-hydrogeological characteristics of the aquifer. Moreover, the hydraulic conductivity values regarding the marbles at the west and metamorphic formations (gneisses, mica-schists) at the N-NE were based on bibliographic references [46–49].

### 4.5. Compilation of Results

The calculations of factors, sub-factors' values, and spatial distribution were performed in ArcPRO® software to extract the digital maps, employing the corresponding equations and attributing class values as referenced in the mentioned Tables above. All data were transformed to raster grids with cell size 100 m × 100 m, which covered the case study area of the Tirnavos basin. The derived maps for each sub-factor, including the weighting factors and the cumulative map for main factors, are depicted in Figures 5–8.

Two vulnerability classes are assigned for the R factor: class I for the greater part of the study area and class II at the central and southeast part of the study area, as shown in Figure 5.

Regarding factor I, all vulnerability classes emerged in the spatial distribution map (Figure 6) following the raster calculations of its three sub-factors. The highest vulnerability class (V) was assigned along the traces of significant fault zones and surface-ground water system interaction zones. The class (IV) occupies the highest area percentage, and its main distribution is at the alluvial basin and the eastern part of the case study area, excluding its central part. The central part of the alluvial basin and most of the karstified domain are characterised by medium vulnerability class (III), while at the north and southeast, the low

class (II) prevails. The lowest vulnerability class (I) has only isolated occurrences at the central and eastern parts of the area.

In the part west of Agia Sofia local community, low- and high-class alternations are presented, while the intermediate medium class is absent, as observed at the I factor distribution map. This fact is probably due to Neogene formations in the specific area and the southern part. However, what differentiates the two parts, is the topographic slope, which is lower in the south than in the west of the Agia Sofia village, where a slope break exists from the mountainous terrain to the alluvial basin. That justifies the appearance of the medium class in the southern part in contrast to the area west of the Agia Sofia village.

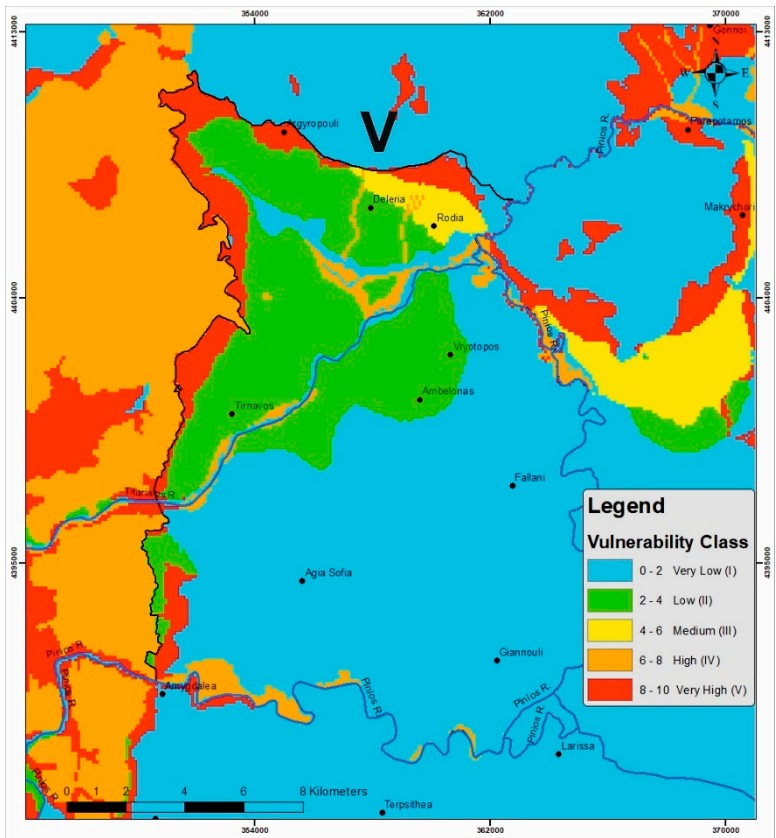

**Figure 7.** Spatial distribution of vulnerability classes according to the V factor in the study area.

In order to compile the V factor distribution map, based on Table S11, the pc values were attributed to a vulnerability class/characterization, corresponding to a V factor value. The map produced after this exercise is illustrated in Figure 7. In this map, the full range of vulnerability classes is also displayed.

Following the approach mentioned in the A factor description paragraph, a value of hydraulic conductivity correlated to vulnerability class was attributed to Table S12. Ultimately, a value of A factor was assigned to each formation and known permeability value point, and subsequently, data values were spatially interpolated with Inverse Distance Weighting (IDW) method to obtain the spatial distribution map for factor A (Figure 8).

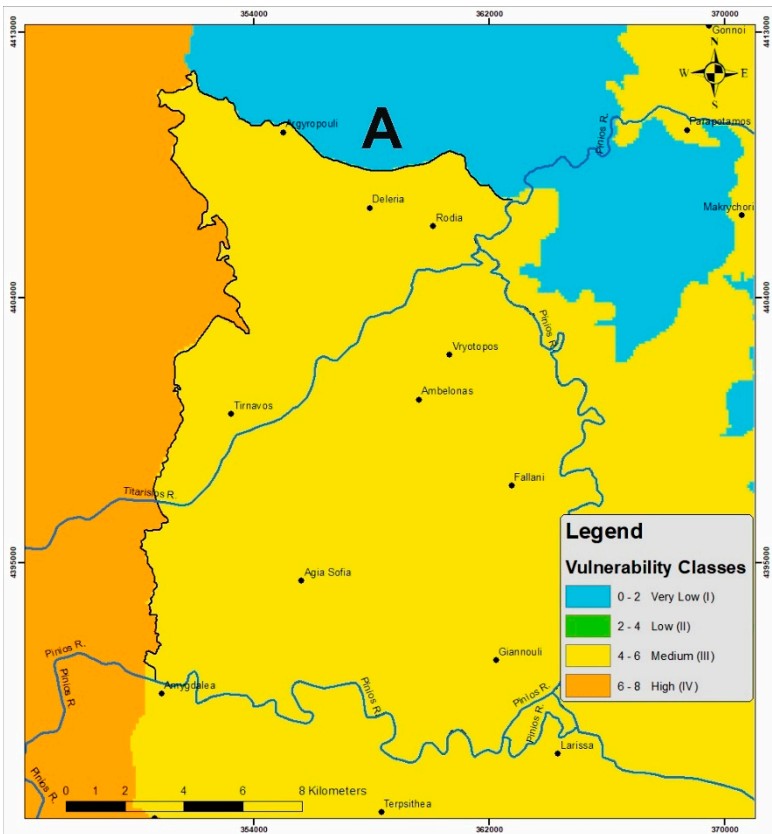

**Figure 8.** Spatial distribution of vulnerability classes according to the A factor in the study area.

Since vulnerability maps were created for all four factors (R,I,V,A), groundwater intrinsic vulnerability (i) was estimated based on Equation (1), considering the corresponding weighting factors. The final calculations between raster images were performed following a normal grid 100 × 100 m cell distribution with ArcPRO GIS software. The value of any cell of the final map was derived by summing up the individual values of corresponding cells of all factor maps, according to Equation (1). The final intrinsic vulnerability map, which was produced following this procedure, is illustrated in Figure 9. Based on this map, the central and southeast parts of the case study area, where alluvial deposits dominate, are characterised by low vulnerability (II), occupying 48.75% of the total area. Comparing this map with the V factor distribution map, it is concluded that protecting the vadose zone in this area is highly effective since factor F is assigned the highest weighting factor. The karstic area at the west and northwest parts of the alluvial basin shows medium (III) to high (IV) vulnerability, occupying 24.04% and 14.14% of the examined domain. This fact reflects the sensitivity of the nature of the formations in these areas, as they are karstified marbles at the west side and conglomerates and talus cone at the northwest margins of the alluvial basin. The dominance of the high (IV) class at the west and north boundaries of the plain areas is reasonable because of the lithology and topography of the existing transition zones. The north and northeast boundaries of the study area appear to have the lowest vulnerability class (I), occupying 13.01% of the total area and reflecting the crystalline bedrock, which is mica-schists and gneisses, respectively. It should be noted that the areas of very high (V) vulnerability occupy a minimum percentage (0.07%) of the total area and are placed on the tectonic structures between Pinios and Titarisios Rivers.

*4.6. Validation*

The validation procedure is critical for the initial vulnerability assessment [27]. The most common approach, particularly for verification of assessments done with overlay and index methods, is to compare the vulnerability map with the actual occurrence of some

common contaminant in groundwater [65,66]. Considering the initial vulnerability concept and the specific characteristics of the study area, validation was performed by comparing the monitored values of nitrates with the modelled vulnerability, as defined by the spatial distribution of the final vulnerability map [26] (Figure 9).

Nitrate values from 43 boreholes were used to create the spatial distribution map of nitrates in the Tirnavos basin, based on previous surveys and sampling campaigns [28,34,36]. Each value is the average result of four sampling campaigns at wet and dry seasons from September 2016–April 2018. These average values of nitrate concentrations were classified into ranges as follows: 0–20 mg/L, class I (very low); 20–40 mg/L, class II (low); 40–50 mg/L, class III (medium); 50–70 mg/L, class IV (high); >70 mg/L, class V (very high). Based on the above classification, which constitutes a modified version of the groundwater nitrates concentration classification proposed in the framework of the Nitrates Directive [67], a spatial distribution map of nitrates' classes was compiled (Figure 10). Nitrates' classes distribution map was subsequently compared to the final vulnerability map, a result of summing RIVA factors (Figure 9). The subtraction of these two maps yields (modelled values—monitored values) the validation map (Figure 11).

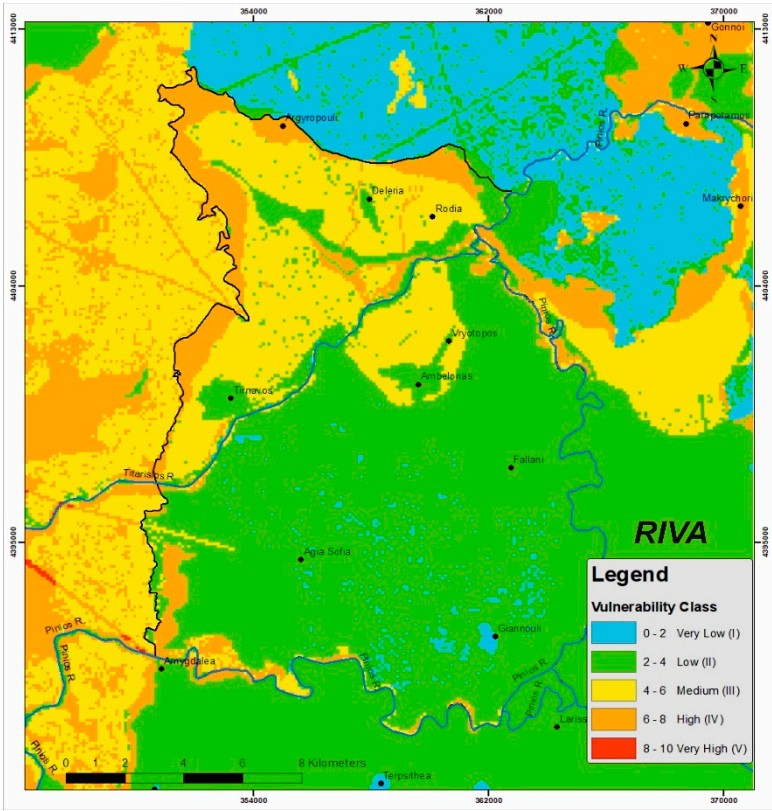

**Figure 9.** Spatial distribution of vulnerability classes according to RIVA method in the study area.

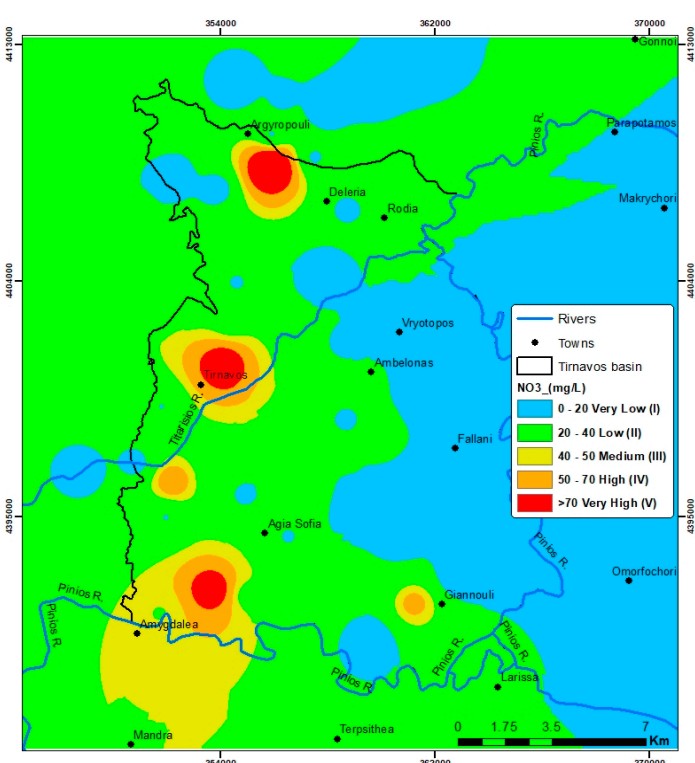

**Figure 10.** Spatial distribution map of nitrates concentrations (average values of four periods, September 2016–April 2018).

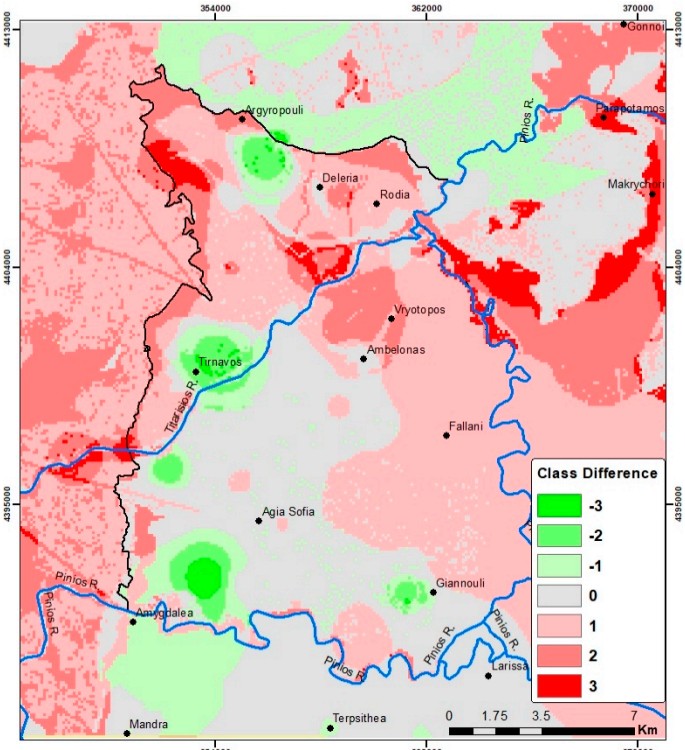

**Figure 11.** Validation map of the RIVA application method in the Tirnavos basin. The class difference between modelled and monitored values, depicted.

In Table 2, the analytical results of the validation procedure are shown. Based on this table, 29.14% of the total area presents a perfect match between modelled and monitored

values (difference class 0). Overall negative difference (underestimating, lower modelled than monitored values) presents 13.20% and positive difference (overestimating, higher modelled than monitored values) 57.65%. According to the original validation of the method [26], a difference of one class (−1 to 1) is accepted as a very good match between the modelled and monitored values. If that percentage is equal to or greater than 80%, then the modelled results are successful compared to the ground truth values.

**Table 2.** Difference of vulnerability class between modelled and monitored values.

| Modelled-Monitored Class | Percentage |
|---|---|
| −3 | 0.33 |
| −2 | 1.61 |
| −1 | 11.27 |
| 0 | 29.14 |
| 1 | 41.11 |
| 2 | 13.82 |
| 3 | 2.72 |
| | 100.00 |

According to the validation performed in the present research, the total area with a difference from −1 to +1 between the modelled and monitored values is 81.52%; thus, the validation may be regarded as successful. Continuing, 96.95% of the total area presents a difference of two classes (−2 to 2), and only 3.05% exhibits a difference of three classes (−3 or +3).

The largest class differences (−3) where RIVA underestimates the potential vulnerability appear in a few individual sites related either to point-source contamination or to the effect of migrating contaminants through lateral crossflows. The latter is probably the case for the deviations in the alluvial basin related to zones of lateral contaminant fluxes and especially so along the southwestern-most edge of the basin. By definition, aquifer vulnerability accounts for the vertical susceptibility of the system and does not account for any lateral crossflows of contaminant plumes from adjacent hydrogeological units. On the other hand, the infiltration of N-containing pollutants from surface water and the transport of nitrate contaminants through soil and groundwater occurs via a series of complex chemical and hydraulic phenomena [68]. As a result of these complex procedures, a horizontal migration of the contaminants is frequently dominant. Hence, practical validation with measured contaminant values at the saturated zone should only be performed at hydrologically "closed" systems without any hydraulic connections with other units (surface or underground); if not, then potential migrations of contaminant plume(s) must be taken into account prior to validation [69].

Indeed, based on the geometry of the groundwater system and its hydrodynamic evolution, as these were discussed in the earlier parts of this work, the area where the highest deviation between modelled and monitored values coincides with the parts of the system where contaminants' migration is influenced by a conurbation of factors where not only percolation is important, but lateral flows do play a significant role. Moreover, it needs to be taken into account that the results of water sampling on nitrate concentrations do not always reflect the regional background regarding contaminant's state, but instead, they may indicate local factors of N-bearing compounds mismanagement. Hence, deviations between the RIVA and nitrates distribution classes are sufficiently explained.

## 5. Discussion

### Insight to the factors used

The effect of precipitation and irrigation on groundwater vulnerability was examined by calculating the R factor in the study area. The mean annual approach is more efficient for strategic planning at a regional scale, whilst the seasonal approach can flag vulnerability aspects that are significant for local or regional scale risk assessment uses [70,71]. Although data from three meteorological stations located at different altitudes were used, both the mean precipitation (P sub-factor) and its intensity (In sub-factor) did not form separate vulnerability classes based on the various ranges that have been defined.

Irrigation (Ir sub-factor) as a factor of vulnerability influence has not been considered in any other method yet. In rural areas where agriculture forms the key socio-economic activity, irrigation is indeed an overlooked parameter of paramount importance in assessing the vulnerability of a groundwater system, as clearly demonstrated through the performed analysis. Especially in the absence of multiple classes for the other 2 sub-factors shaping R factor (P + In), the irrigation sub-factor considerably influenced the spatial distribution of the R factor. Given that the average field size at the study area is rather small (ca 1 ha), it is thought that should irrigation system data occur at the field scale, a more detailed spatially diversified distribution of the R factor would have been yielded. In turn, such an approach would provide a higher spatial resolution assessment of the system's vulnerability concerning this particular sub-factor. On a regional scale, however, this differentiation would have not led to considerably different results apart from the potential for even better agreement between the vulnerability map and the ground-truth nitrate concentrations and, in particular, the ability to better explain hot spots of the nitrates concentration distribution map. However, in the absence of such detailed data, the dominant irrigation system per local irrigation organisation inherently incorporates the assumption of the least water-efficient irrigation system used. Hence, the higher irrigation water is applied on the field, which subsequently leads to a relative over-estimation of water use and, therefore, higher assessments of vulnerability due to this sub-factor. Overall, the approach followed results in a more conservative estimate of vulnerability, i.e., a more environmentally sound approach.

Assessing I factor incorporates evaluating the combined action of slope and vegetation, which constitutes a straightforward procedure based on the principle that the lower the slope and the denser the vegetation, the greater the class of attributed vulnerability. It also includes evaluating permeability of the top 1.5 m of soils and consolidated geological formations, as assigned with the relevant tables developed by the method. Last, it incorporates the effect of linear features, including tectonic lines [72,73] and hydrologically interacting river stretches [74,75]. As a whole, the subjectivity of this factor is rather limited due to the wide classes distinguished for the evaluation of each sub-factor and the overall sound principles it adopts in the definition of the considered sub-factors.

As already stressed, V factor constitutes the most influential factor in assessing the examined system's vulnerability, and this is acknowledged by assigning to the final vulnerability index calculation formula a very high weighting factor (a = 0.4). Hence, it is rather critical to evaluate this factor carefully and based on reliable information to avoid misconceptions on the final spatially distributed vulnerability product. Out of the three considered sub-factors, the dominant lithological type (ly) and the corresponding fracturing or karstification degree (f), which represent the bulk permeability values per formation and the alterations imposed due to secondary deformations, are safely assessed utilizing bibliographic references [28,61,63], geological maps [30,31], field observations [28,34,36], and the adopted classification scheme of RIVA, as earlier presented [26]. Calculation of the protective cap, which forms the third sub-factor (pc), is the most critical and presents the highest risk for erroneous calculations, especially when groundwater system vulnerability assessments are attempted for complex geometry environments, as the examined one is. However, the adopted approach and utilization of the detailed data available on the system's geometry for most of its parts ensures that performed assessments are reliable and, in fact, shifted towards the conservative margin of analysis. Thus, calculations tend to

accept the minimum thickness of protecting cap in the few parts of the basin where relevant data for this sub-factor are not detailed. As with the I factor, this provides a safety margin in favour of environmental protection, assuming the worst-case scenario on the geometry of the considered system.

Last, factor A reflects the influence of hydraulic conductivity on intrinsic vulnerability, and its spatial distribution is deduced based on pumping test analyses, augmented by region-specific and more generic but well-established bibliographic references [46–49] transposed to wide range classifications, as earlier presented. Hence, the margin of error in assessing the spatial distribution of this factor is limited.

Validation of the method was performed on the basis of a considerable population of real monitoring data that cover dry and wet hydrological conditions for two years and are typical of an agriculture-related pollutant. Considering the particularities of the studied region as it is reflected by geological structure, hydrogeological setup, and regional hydrodynamic evolution mechanisms, validation results are satisfactory, demonstrating the validity and efficacy of RIVA in similar complex environments. Spatially distributed zones of different vulnerability classes are delineated accurately and reliably even though the tested hydrogeological environment is characterised by a high degree of complexity.

Observed deviations relate mainly to intrinsic vulnerability assessment methods' inherent limitations in accounting for horizontal flow driven migration of pollutants. Moreover, it is expected that the produced results would have been further improved if monitored data utilized for validation were originating from a single aquifer of the examined system rather than reflecting an average concentration of the entire system. That is the result of the construction characteristics of most production wells in the studied region, which tap the entire groundwater system that consists of multiple aquifers, the hydrochemical evolution of each one of which is controlled by different mechanisms. Last, the highest deviations noted reflect hot spots areas that are most likely related to one or more of the following reasons: (a) N-compounds mismanagement at the vicinity of the well heads, (b) systematic over-irrigation leading to excessive leaching of contaminants, and (c) poor construction characteristics of production wells that enable migration of contaminants to deeper hydrogeological strata through the wells' gravelpack.

In total, implementation of RIVA in the studied region provides a valuable tool for regional planning and management of natural resources that the area lacks, leading to considerable deterioration of existing groundwater reserves. Compared to other approaches, the selected method enables uniform consideration and evaluation of the entire region structured of highly contrasting character lithological typologies, rendering vulnerability assessment easier and comparable amongst the existing geological media. Of particular importance is the consideration of irrigation water management, which leads to a more comprehensive assessment of the essence in rural areas where intensified agriculture is practised. Even though irrigation is an externality, being an established condition, it directly influences vulnerability and should be given a thorough, reliable, and pragmatic assessment. In several parts of the world, data scarcity is a fact. Still, reliable vulnerability assessments are imperative and may not await appropriate state-of-the-art data collection. RIVA is not a data-intensive method that can easily retrieve information based on sound geological and hydrogeological knowledge. Obviously, enhanced results may be obtained, and a higher spatial resolution achieved should spatially dense and reliable data-driven calculations be employed.

**Challenges and problems**

Groundwater systems vulnerability assessment is a valuable tool with profound applications in environmental science. One of them could potentially be its incorporation, as a factor in the delineation of Nitrate Vulnerable Zones (NVZs), in the framework of the Nitrates Directive [67]. Thessaly as a whole is one of the first regions of the country to be declared as an NVZ in the late 1990s [76], using as key criteria the surface and groundwater concentration of nitrates, the N-input, and the geomorphological characteristics of the region. Groundwater system vulnerability consideration would add to the reliability of the

NVZ assessment and perhaps even lead to a considerably differentiated delineation pattern, accounting for the essential properties that control the potential leaching of contaminants. Therefore, such an approach would lead to a more comprehensive and meaningful delineation of NVZs and greatly assist the design of appropriate and targeted, thus efficient, measures as part of the Action Plan in the framework of the Nitrates Directive. Such an approach is directly in line with the Common Agricultural Policy, which identifies the direct relationship between agriculture and groundwater resources and the need for effective protection and preservation of the latter from agricultural inputs [77,78].

Spatially distributed flow models have been developed for the studied area and the wider Thessaly region [79–81], which provide a reliable simulation of the flow domain accounting for the key hydrodynamic evolution mechanisms. Contamination transport and even hydrogeochemical modelling coupled with such flow models would enhance understanding of evolution mechanisms and improve validation of the results. Advective and dispersive migration elements of the pollutants could be quantified and isolated, thus enabling backwards analysis of nitrates concentration distribution deduced from the monitoring exercise, accounting for reducing or oxidizing reactions that influence groundwater chemistry and thus the nitrates concentrations in the collected and analysed water samples. In this way, leaching would be the sole pathway to be reflected on the nitrates distribution, thus directly relating to the evaluation provided by RIVA, which assesses vulnerability considering the vertical pathways of potential pollutants. On a different viewpoint, such models could be coupled to RIVA to account for the advective and dispersive migration of a potential pollutant, thus contributing to a more holistic assessment of groundwater systems' vulnerability. In such an approach, the vulnerability would be calculated as a function of the examined parameters in the framework of RIVA (for leaching potential assessments) and lateral crossflows from surrounding formations or groundwater systems.

As pointed out in previous parts of the paper, seasonal variations of precipitation patterns lead to differentiated calculations of key factors of the vulnerability index. Likewise, consideration of seasonal groundwater levels as opposed to mean annual or inter-annual levels may lead to considerable differences in the calculations of the protective cap sub-factor of V factor, which has the highest weighting factor in the calculation of the vulnerability index. Previous studies have demonstrated this in similar geo-climatic environments [82,83]. In the framework of climate change, considerable variations of key climate parameters, including precipitation depth and intensity, temperature, and evapotranspiration, are anticipated, especially for the region Thessaly, which is considered amongst the vulnerable regions along with the entire Mediterranean [84–86]. These changes will potentially affect the hydrological balance of the system in terms of anticipated natural recharge and abstractions given that irrigation water needs are expected to increase under the business as usual scenario. In turn, depth to groundwater is expected to be affected. Prolonged droughts duration and frequency along with the already experienced severe floods because of high rainfall intensity directly differentiate directly the values of several sub-factors considered in the vulnerability assessment under the RIVA method.

Likewise, improving irrigation water efficiency is also expected in response to climate change and resilience development. Finally, crop distribution changes are already being discussed as an adaptation measure. These are some of the key changes expected in the future that will influence the vulnerability of the considered groundwater system. The factors above will alter the spatial distribution of vulnerability. However, some of them will increase it, whilst others decrease it. Therefore, the overall prediction of the anticipated outcome may not be safely made empirically. Still, reliable assessments of the forecasted vulnerability under climate change environment are important to augment measures designed to increase groundwater systems' resilience and overall water sector safety.

## 6. Conclusions

This study assessed the intrinsic vulnerability of the Tirnavos basin (Central Greece) groundwater system using the RIVA method. RIVA has the advantage that it can be applied to all types of groundwater bodies independently of the specific conditions, lithologic phases, and aquifer typology of each area. Four main factors were used to represent the natural hydrogeological conditions of the specific area: recharge, infiltration, vadose zone, and aquifer.

Modelled results were validated with ground-truth values of nitrates obtained from 43 wells and proved to be quite successful, as they presented over 80% of matching (negligible or small deviations between modelled and monitored values). The few deviations are attributed to inherent uncertainty factors, such as the interpolation of the various factors used and the lateral contamination transport, which the index and overlay methods cannot assess without the help of a spatially distributed model.

The outcomes of the RIVA application to the Tirnavos basin can be further exploited as a preliminary tool for decision making and strategic assessment. At a regional scale, the results may be further valorised for the delineation of NVZs, considered as a key target for the trade-off between sustainable agriculture and water resources protection.

**Supplementary Materials:** The following supporting information can be downloaded at: https://www.mdpi.com/article/10.3390/w14040534/s1, Table S1: Correlation between annual precipitation (mm/y), P values, and vulnerability classes; Table S2: Correlation between rainfall intensity (mm/d), In values, and vulnerability classes; Table S3: Correlation between irrigation dose, In values, and vulnerability class; Table S4: Classification of topographic slope (s) and correlation with vulnerability classes; Table S5: Categorization of vegetation according to CORINE land-use classification (EEA, 2020); Table S6: Calculation of F values according to soil (s) and vegetation (v) parameters and link with vulnerability classes; Table S7: Calculation of S values for the soils and correlation with the vulnerability class (USDA, 1999); Table S8: Calculation of S values for consolidated geological formations and correlation with permeability and the vulnerability class (Lewis et al., 2006); Table S9: Classification of layer reference values (ly) for representative geological formations; Table S10: "f" factor values according to the assessed fracturing or karstification degree; Table S11: Protective cover (pc) values and corresponding vulnerability characterization and V factor values; Table S12: The suggested link between hydraulic conductivity and A factor values, corresponding to specific vulnerability class.

**Author Contributions:** I.V. conceived the methodological approach, performed the data processing and supervised the drafting and revision of the final text. E.T., A.P. and G.S. participated pin data processing, verified and optimised the methodology, contributed to the discussion parts, and revised the final text. All authors have read and agreed to the published version of the manuscript.

**Funding:** This research received no external funding.

**Institutional Review Board Statement:** Not applicable.

**Informed Consent Statement:** Informed consent was obtained from all subjects involved in the study.

**Data Availability Statement:** Data is contained within the article.

**Conflicts of Interest:** The authors declare no conflict of interest.

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
