# Peer review of "Groundwater Vulnerability Analysis of Tirnavos Basin, Central Greece: An Application of RIVA Method"

_water, doi:10.3390/w14040534_

Round 1

Reviewer 1 Report

An index-based method RIVA was used for the groundwater vulnerability analysis of an area in central Greece. The vulnerability map was validated with ground-truth values of nitrates obtained from boreholes and proved to be quite successful, with mostly only small deviations between modeled and monitored values. Although the methodology used is not new, the information and interpretations collected are definitely interesting and deserve publication in this journal. I have only a few minor comments, which are listed below.

  1. Page 4, Line 117: Delete “Location” in the caption of Figure 2.
  2. Page 4, Lines 122 and 131-132: Since two springs are mentioned (one with 2 different names), I suggest marking them on the map.
  3. Page 4, Line 131: Correct “Figure 4” to “Figure 2”.
  4. Page 4, Line 137: Perhaps the references should be 33, 34?
  5. Page 6, Line 181: Delete “Land” in the caption of Figure 4.
  6. Page 10, Line 388: The reference to Table 1 seems wrong to me.
  7. Page 23, Table A6: It is not clear to me what (V) means.

Reviewer 2 Report

The groundwater vulnerability assessments is important for the risk assessment and decision-making of groundwater resource management.This manuscript presents a case study the intrinsic vulnerability assessment of the Tirnavos basin (Central Greece) groundwater system using the RIVA method. RIVA method is proved as a fair trade-off between succeeded accuracy, data intensity and investment to reach highly accurate results for the susceptibility assessment to surface-released contamination. The manuscript was well written, however, there are some minor flaws which need to be addressed before publication.

  1. L175: The contribution of each factor in equation 1 is not equivalent due to the different degrees of importance. The weighting factors (a,b,c,d), which were calculated with the use of Analytic Hierarchy Process (AHP) and proposed by [26]. Are the weighting factors (a,b,c,d) are directly used the factors from reference 26 or adjusted according to the situation of the study area? More discussion of the determination of the weighting factor is suggested to be added.
  2. What are the most important factors for the groundwater vulnerability assessments of the study area of the Tirnavos basin? It is better to add more description and discussion.
  3. The legends of the Fig. 5 P ,In , Ir and Fig.6 F,S,C are not clear enough.

Reviewer 3 Report

This manuscript was organized well. I think it can be accepted as such.
